# Human Activity Recognition Using Attention-Mechanism-Based Deep Learning Feature Combination

**DOI:** 10.3390/s23125715

**Published:** 2023-06-19

**Authors:** Morsheda Akter, Shafew Ansary, Md. Al-Masrur Khan, Dongwan Kim

**Affiliations:** 1Department of Electronics Engineering, Dong-A University, Busan 49315, Republic of Korea; 2Department of ICT Integrated Ocean Smart and Cities Engineering, Dong-A University, Busan 49315, Republic of Korea

**Keywords:** attention mechanism, deep learning, feature combination, human action recognition

## Abstract

Human activity recognition (HAR) performs a vital function in various fields, including healthcare, rehabilitation, elder care, and monitoring. Researchers are using mobile sensor data (i.e., accelerometer, gyroscope) by adapting various machine learning (ML) or deep learning (DL) networks. The advent of DL has enabled automatic high-level feature extraction, which has been effectively leveraged to optimize the performance of HAR systems. In addition, the application of deep-learning techniques has demonstrated success in sensor-based HAR across diverse domains. In this study, a novel methodology for HAR was introduced, which utilizes convolutional neural networks (CNNs). The proposed approach combines features from multiple convolutional stages to generate a more comprehensive feature representation, and an attention mechanism was incorporated to extract more refined features, further enhancing the accuracy of the model. The novelty of this study lies in the integration of feature combinations from multiple stages as well as in proposing a generalized model structure with CBAM modules. This leads to a more informative and effective feature extraction technique by feeding the model with more information in every block operation. This research used spectrograms of the raw signals instead of extracting hand-crafted features through intricate signal processing techniques. The developed model has been assessed on three datasets, including KU-HAR, UCI-HAR, and WISDM datasets. The experimental findings showed that the classification accuracies of the suggested technique on the KU-HAR, UCI-HAR, and WISDM datasets were 96.86%, 93.48%, and 93.89%, respectively. The other evaluation criteria also demonstrate that the proposed methodology is comprehensive and competent compared to previous works.

## 1. Introduction

The process of identifying distinct human actions and activities/motions and providing an appropriate response is referred to as human activity recognition (HAR). The procedure starts with the use of sensors that can capture motions, gestures, and other behaviors which are discerned as human motion. These action commands are then transformed from the movement data into HAR code that computers can interpret and execute. The Internet of things (IoT), cloud, and edge computing are some of the interconnected sensing technologies that have seen significant developments in the previous ten years. Wearable sensors that are widely employed in IoT applications can be used to quickly record different body movements for human activity detection. Wearable inertial measurement unit (IMU) sensors, which are made up of accelerometers and gyroscopes, have developed quickly in recent years, making it simple to detect and track human movement. This has made it possible for HAR to be used in a variety of industries, including healthcare, biometrics, and also other fields such as human emotion recognition [1]. The significance of HAR based on wearable sensors is made clear by the fact that it is not only restricted to exercise-related activities but can also be used for classifying and documenting various daily behaviors such as eating, drinking, brushing teeth, and detecting sleep irregularities. HAR has been extensively researched, starting from handcrafted feature-based approaches to leveraging advanced deep learning (DL) techniques. Earlier HAR techniques primarily relied on manual feature extraction and machine learning (ML). Handcrafted feature extractors such as HOG [2], SIFT [3], SURF [4], etc., were utilized for extracting low-level features. Some examples of early handcrafted feature extractors from various research are discussed below.

Chen et al. [5] proposed GFSFAN, a two-stage genetic-algorithm-based feature selection method for HAR. The algorithm demonstrates high classification performance with a compact feature subset extracted from raw time series of nine activities of daily living. Six classifiers are used to evaluate the impact of feature subsets selected by different feature selection algorithms on (HAR) performance. Additionally, the introduction of sensor-to-segment calibration and lower-limb joint angle estimation improves HAR performance. Shi et al. introduced [6] a novel algorithm known as standard deviation trend analysis (STD-TA) for the recognition of transition activities. In contrast to other existing methods that directly categorize transitions as basic activities, this approach offers improved overall performance. The STD-TA algorithm achieves an accuracy exceeding 80% when applied to real data, indicating its efficacy in accurately identifying transition activities. Bay et al. introduced [3] the SURF interest point detector and descriptor, a unique scale and rotation-invariant approach. Because the Hessian matrix performed well in terms of computation speed and precision, it was chosen as the foundation for their detector. The authors provided a quick and effective interest point detection–description system that outperformed the current methods regarding both speed and accuracy. Despite the wide use of feature-based approaches in HAR, they struggled with noisy time series signal data and datasets with numerous features. In contrast, the convolutional layers in CNNs excel at handling noise and accommodating multiple features, resulting in superior performance compared to traditional methods. However, it is worth noting that some traditional hand-crafted methods, as exemplified by research in [7,8], can still outperform modern approaches in specific cases, highlighting their ongoing relevance and potential for achieving higher accuracy. Moreover, Ref. [9] describes the more improved hierarchical hidden Markov model (HMMM) in terms of sensor-based activity recognition, and Ref. [10] describes the latest platform and interactive machine learning solution in hand-crafted feature-based HAR in recent research.

The limitations of these approaches prompted researchers to explore alternative methods in HAR. This led to the emergence of model-based approaches, supervised ML algorithms, and various state-of-the-art techniques. These newer methods emphasized the utilization of sensor-based data rather than image-based data in HAR. Yiming et al. proposed [11] HDESEN, a novel method for smartphone sensor-based HAR. It employs selective ensemble learning with differentiated ELMs, incorporating hybrid diversity enhancement and IBGSO for subset selection. Statistical features are extracted, and diverse base models are constructed using distinct training data subsets and random and optimized subspaces. A pruning method based on GSO selects the optimal sub-ensemble. HDESEN outperforms existing approaches on UCI-HAR and WISDM datasets, improving HAR performance. Yin et al. proposed [12] a triplet cross-dimension attention method for sensor-based activity recognition. It utilizes three attention branches to capture cross-interactions between sensor, temporal, and channel dimensions. Extensive experiments on multiple datasets validate its effectiveness and demonstrate improved classification performance across different backbone models. Visualization analysis supports the findings, and the implementation is evaluated on a Raspberry Pi platform.

In recent years, researchers have increasingly adopted DL models for HAR due to their impressive performance and advantages over traditional methods. The accuracy of sensor-based HAR has also improved significantly with the emergence of DL technologies. Recurrent neural networks (RNNs) and convolutional neural networks (CNNs) have emerged as crucial techniques in sensor-based HAR and have shown significant superiority over traditional approaches. Among all the research, Lokesh et al. proposed [13] an intelligent HAR system using CNN and LSTM. The model achieved up to 97.5% accuracy on the WISDM dataset with a low-complexity architecture and 21,922 trainable parameters. Results were visualized using a convolutional matrix and learning curves, showing promising performance across diverse environments and domain structures. Ponnipa et al. proposed [14] a sensor-based HAR system using the InceptTime network. The model achieved an accuracy of 88% on the PAMAP2 dataset, outperforming other baseline DL models. Haiyong et al. proposed [15] DSFNet, a DL model for multi-sensor fusion. It leverages acceleration sequences for motion trends and uses a Transformer encoder for feature extraction. Angular velocity is utilized for pose direction and velocity. Decision-level fusion enhances performance, outperforming existing methods on the CZU-MHAD dataset. Qi et al. proposed [16] RepHAR, a resource-efficient dynamic network for low-cost hardware-constrained HAR. By combining multibranch topologies and structured reparameterization, RepHAR achieves a trade-off between speed and accuracy. Experimental results show improved accuracy (0.83–2.18%) and parameter reduction (12–44%) on multiple datasets. RepHAR also outperforms the MB CNN model by running 72% faster on Raspberry Pi. Challa et al. [17] introduced a multibranch CNN-BiLSTM model that captures features with minimal data pre-processing. The model can learn both local features and long-term dependencies in sequential data by using different filter sizes, enhancing the feature extraction process.

The discussion shows there is still room for improvement in the current models. Among the modules, an attention mechanism, an atrous spatial pyramid pooling module, a spatial-channel squeeze, and an excitation module can be added to the CNN model in the field of visual target recognition to preserve significant features and suppress inconsequential features, and it has been shown to be useful in elevating the target identification rate. Moreover, the discussed literature still has room for enhancement both in terms of recorded performance and robustness, reliability, and efficacy across numerous datasets, especially the KU-HAR dataset. Considering these research scopes, this study proposes a new HAR classification method using a functional DL model having 250,798 training parameters integrated with an attention mechanism. The method has been designed as a generalized model structure with CBAM modules so that the same model structure works for different datasets with small tuning. This method aims to extract crucial characteristics from sensor data and achieve higher classification accuracy compared to existing models. The proposed model combines extracted features after the third and fifth phases, enhancing feature characteristics and improving overall accuracy. Additionally, this model is suitable for use in IoT and low-compute-power devices, as it does not require substantial filtering, noise removal, additional signal preprocessing, or manual feature extraction methods. The proposed model integrates the attention mechanism with DL and combines the extracted features within the DL model, referred to as attention-mechanism-based deep learning feature combination (AM-DLFC), which is the novelty of this research. Three publicly available datasets, KU-HAR, UCI-HAR, and WISDM, were used to evaluate the proposed model. The results demonstrate that the proposed method achieves a high accuracy rate of 96.86% on the KU-HAR dataset, and the other evaluation metrics also indicate the robustness and reliability of the proposed method over other compared works.

Therefore, this model is lightweight and easy to train and use. The following are the contributions of this research:Developing an accurate and lightweight 2D CNN model with minimal pre-processing that works for multiple datasets with minimal tuning.Achieving higher accuracy on the primary dataset and comparing them with the existing models’ results.Testing the robustness of the proposed model by analyzing two other benchmark datasets and comparing the results with other existing methods.

The remainder of this article is structured as follows: Section 2 depicts the utilized methodology of this work; Section 3 presents the experiments and results. Finally, this paper is concluded in Section 4.

## 2. Methodology

This study presents a novel framework for the classification of human activities. The proposed methodology is illustrated in Figure 1, outlining the overall approach. Three publicly available datasets were utilized for both training and testing the model, and Figure 1 serves as an illustration of their common methodology. During the pre-processing stage, minimal adjustments were made, primarily involving the construction of spectrograms for the 2D CNN model. This pre-processing procedure was uniformly applied to all three datasets, and the specific steps were elaborated upon subsequently. Following the pre-processing stage, the data was employed for training and testing the DL model, which is further elucidated in greater detail below.

### 2.1. Constructing Spectrograms from HAR Samples

In many instances, 2D CNNS outperform 1D CNNs with the same number of parameters. To take advantage of 2D convolution and 2D pooling layers, this work converted the 1D time-series signals illustrated in Figure 2 into 2D images.

To convert the signals into 2D images, this work utilized the technique of spectrogram. In the case of the KU-HAR and UCI-HAR datasets, the spectrogram conversion process involved both 3-axis accelerometer and 3-axis gyroscope data. However, for the WISDM dataset, only the 3-axis accelerometer data was utilized for the generation of the spectrograms. A spectrogram shows the signal strength over time at a different frequency contained in a specific waveform illustrated in Figure 3.

This work considered short-time Fourier transform (SIFT) for generating the spectrogram. If the considered HAR sample in this work is *x*[*n*] and the window function is *w*[*n*], then the converted spectrogram is as follows:(1)$$x\left(m,w\right)=\sum_{n}x[n]\;w\left[n-m\right]\;exp\left(-jwn\right)$$

Here, *m* is the time index; *w* is the frequency index; and *exp*(−*jwn*) is the complex exponential function, where *j* is the imaginary unit, and w is the frequency index.

The samples of the three datasets each have a distinct size. Each sample in KU-HAR has 300 features, or 3 s worth of data, while each sample in UCI-HAR has 128 features, or 2.56 s worth of data. The samples from the WISDM dataset were collected at a lower rate than those from the other two. This research chose to maintain a constant sample duration. Therefore, the utilized approach manually segmented the dataset’s raw data using a non-overlapping 3-s window. This process resulted in 20,750, 10,299, and 18,455 samples for the KU-HAR, UCI-HAR, and WISDM datasets, respectively. After converting to the spectrogram, the raw samples turned into a shape of 8 × 129 × 6, where the height, width, and channel of the samples are 8, 129, and 6, respectively.

### 2.2. Proposed Model Architecture

In this paper, a 2D CNN-based DL model for the classification of HAR is proposed, and functional API was used to develop the DL model since functional API offers more flexibility and control over the layers than the sequential API. Furthermore, the DL model was reoriented by integrating a convolutional block attention module (CBAM) for refining the features to obtain better results. Figure 4 displays the architecture of the AM-DLFC model to classify human activities. The design of the developed AM-DLFC model consists of 6 convolutional blocks, where the 1st convolutional block takes the input with the shape of 8 × 129 × 6.

Each convolutional block contains a 2D convolutional layer along with a CBAM module. The basic configuration of the CBAM module is demonstrated in Figure 5. To prevent the model from becoming overfit on the training data, certain dropout layers were also utilized, which improved the model’s performance on the testing samples. The filter sizes that were used in the convolutional layers of the proposed model are 16, 32, 32, 64, 64, and 128, respectively, for each layer. The CBAM blocks are stacked immediately after the convolutional layer in each convolutional block. In this work, a CBAM module was integrated, which is a general, lightweight, and trainable module, and it can be seamlessly integrated into any CNN architecture with very little overhead.

Every convolutional block in deep networks employed CBAM to obtain successive “Refined Feature Maps” from the “Input Intermediate Feature Maps”. These two sub-modules are named the channel attention module (CAM) and the spatial attention module (SAM). By using a 1D CAM, F CAM ∈ ℝ 1 × 1× 1, the CBAM module extracts the crucial channels from the intermediate feature map A and creates a channel-refined feature map B. Mathematically,
B = F CAM (A) ⊗ A

Then the CBAM module employs a 2D SAM, F SAM ∈ ℝ 1 × H × W at the feature map B and finally produces a spatial attention map.
C = F SAM (B) ⊗ B

The first convolutional block also contains a max-pooling layer to reduce the spatial dimension after the CBAM block. Therefore, the output dimension of the first block was reduced from 8 × 129 × 6 to 4 × 64 × 16. Then after moving to the second block, the input shape was 4 × 64 × 32 as the filter size was 32 in the second convolutional block.

A max-pooling layer was not used in the second block since it would have further reduced the overall dimension size, which, in turn, might also dissipate the feature after using a few convolutional blocks. The structure of the third convolutional block is also like the second convolutional block, and max pooling was also not used in the third convolutional block. After that, the third and second convolutional block’s outputs were added to make the features more robust. To reduce the spatial dimension, a max-pooling layer was used, which resulted in an output shape of 2 × 32 × 32. The internal layers of blocks four and five were designed exactly as blocks two and three; however, different filter sizes were used. Later the output of blocks four and five was also added, and the resulting output shape was 2 × 32 × 64. Then again, another max-pooling layer was used to reduce the spatial dimension further, and the shape was 1 × 16 × 64. Finally, with a filter size of 128, the output was added to another convolutional block with a resulting shape of 1 × 16 × 128.

After completing the feature extraction phase, the features were flattened. Later, 5 densely connected layers with neuron numbers of 128, 64, 48, and 32 were used, respectively. Finally, the output layer was used, with 18 neurons for the KU-HAR dataset and 6 neurons for UCI-HAR and WISDM datasets, due to there being 18 classes for the KU-HAR dataset and 6 classes for the UCI-HAR and WISDM datasets, respectively. 

## 3. Experimental Procedure and Results

### 3.1. Dataset Description

Three different benchmark datasets were employed to collect time-domain human activity samples for this study. The University of California Irvine HAR (UCI-HAR), Wireless Sensor Data Mining (WIDSM), and Khulna University HAR (KU-HAR) provided these data sets. In the following subsections, a brief explanation of how these datasets were generated, their substance, and some of the most recent publications that have been authored about them are discussed.

#### 3.1.1. KU-HAR Dataset

This dataset, which consists of 1945 time-domain samples that span into 18 distinct activity classifications, was released in 2021 by Nahid et al. [18]. The activities of this dataset are shown in Table 1.

For ML tasks, the samples were subsequently divided into 20,750 separate, non-overlapping subsamples. A total of 90 participants, aged between 18 and 34, had smartphones fitted around their waists to record the data. The outputs from a tri-axial accelerometer and gyroscope were recorded and compiled at 100 Hz sampling frequency. During data collection, the acceleration generated by gravity was omitted, and no noise filtering process was conducted on the activity data. For optimum customizability, the given dataset combines the raw samples as well as the segmented subsamples. The dataset is described in depth in [19], along with comparisons to other datasets such as UCI-HAR and WISDM. The subsamples and RF classifier were domain-transformed in a fundamental classification framework by the authors as part of their activity recognition research.

#### 3.1.2. UCI-HAR Dataset

The UCI-HAR dataset [20,21] is considered one of the popular datasets used for benchmarking activity recognition, which was produced by Anguita et al. It includes 10,299 instances of six different actions that were gathered with the aid of smartphone built-in sensors.

A total of 30 people, whose ages ranged from 19 to 48, provided the samples. Readings from a wrist-mounted smartphone’s accelerometer and gyroscope were taken at a constant frequency of 50 Hz. After being preprocessed to reduce noise, the samples were segmented with 2.56 s sliding window and a 50% intersection between two subsequent samples. The individual activity samples contain nine distinct categories of data, which are denoted as channels. These movement data contain the total acceleration along the three axes, the triaxial angular velocity, and the triaxial body acceleration. There are 128 readings of the equivalent activity for each channel. The classes and samples from this dataset are described in detail, together with visualizations, in [22]. Table 2 contains the activities of the UCI-HAR dataset. The UCI-HAR dataset is commonly utilized in this field of study. As a result, a lot of research on the topic has been published, each with novel, refined ML models and opposing findings. Ronald et al. [23] created iSPLInception to categorize the classes of four HAR datasets, incorporating UCI-HAR. Bhuiyan et al. [24] created a modified bag-of-words model and utilized k-nearest neighbors to categorize these activities. To capture features from the activity signals’ multiple altered spaces, Mahmud et al. [25] proposed utilizing a multi-stage training approach using CNN. The developed technique was implemented on three HAR datasets. The architecture suggested by Ghate and Hemalatha uses three 1D CNNs to extract features and random forest (RF) for activity categorization [26].

#### 3.1.3. WISDM Dataset

This dataset was released in 2012 and was developed by Kwapisz et al. [27,28]. It incorporates six classes of action signals that were obtained from 29 participants. In contrast to UCI-HAR, the WISDM dataset only contains the tri-axial accelerometer data. Compared to most HAR datasets, the data were collected at a 20 Hz sampling frequency. The dataset consists of 1,098,207 time-domain data that were then converted to 5424 samples. Table 3 contains the activities of the WISDM dataset.

The HAR method developed by Peppas et al. [29] combines forty statistical parameters extracted from the WIDSM dataset with the convolutional properties of a two-layer CNN. Jalal et al. [30] integrated the auditory, statistical, and frequency data before applying a genetic algorithm to choose the features. The developed HAR approach was evaluated using a sample of the WISDM dataset. Nafea et al. coupled [31] CNNs with BiLSTM that had varied kernel dimensions to extract features from the samples at various resolutions. Additionally, UCI-HAR and WISDM, which are based on global average pooling (GAP) layers and are a component of an LSTM-CNN-based approach, were described by Xia et al. in their study [32]. Several CNNs were used with gated recurrent units (GRUs) as part of a system for activity recognition that Dua et al. proposed [33]. Their methodology was examined using the UCI-HAR and WISDM datasets.

### 3.2. Implementation Details

The DL framework that was utilized was Keras version 2.6.0, and the development language used was Python version 3.6.13. The Adam optimizer was employed, the mean squared error loss function, a batch size of 32, a learning rate of 1 × 10^−4^, and a total of 200 epochs throughout the training of the AM-DLFC model. The experiments and model training were conducted on a machine running Windows 10 with 32 GB of RAM, an Intel Core i9-7900x processor clocked at 4.30 GHz, and an NVIDIA Geforce RTX 2080Ti graphics card.

This section describes the experimental setup and the results that the designed classification model generated on the three datasets examined in this study. The model’s effectiveness will be assessed using common assessment measures such as classification accuracy, loss, F1 score, and confusion matrix. One of the more obvious measures for evaluating all the correctly detected instances is accuracy. Since all the classes are equally essential, it is widely employed.
Accuracy Score = (True positive + True negative)/(True positive + False negative+ True negative + False positive)

On the other hand, the discrepancy between the predicted values by the model and the actual values of the problem represents the loss. The F1-score computes the subcontrary mean of both precision and recall, which provides a more reliable evaluation of misclassified instances compared to the accuracy metric.
F1 Score = 2 × Precision Score × Recall Score/(Precision Score + Recall Score)

A matrix with dimensions of N × N, known as the confusion matrix, is employed to assess the efficiency of a model, where N denotes the total number of classes being predicted.

To address the supervised learning issue, the samples were randomly split into training and testing subgroups, accounting for 75% and 25% of the entire sample accordingly.

### 3.3. Results on KU-HAR Dataset

The dataset that was used was divided randomly in a ratio of 7.5:2.5 to train and test the model. The outcome shows a classification accuracy in both the training and testing phases of over 96.86%. The classification accuracy on both training samples and test samples up to 200 epochs is shown in Figure 6a. The figure illustrates that the precision of training began at the 32nd epoch, and it then experienced a very slight fluctuation on the 85th epoch. The minor fluctuation was quickly overcome, and the curves returned to being stable. Finally, the model ran very smoothly up to the 200th epoch. A peak accuracy of 96.99% was achieved on the 183rd epoch. The loss function and F1 measurement are two frequently utilized metrics to measure the proficiency of classifiers. Figure 7a and Figure 6b describe the loss curve and F1 curve, respectively, at each epoch of the proposed AM-DLFC model. The F1 measurement curves in Figure 6b substantially resemble the accuracy curves in Figure 6a, demonstrating that the classification results are accurate and not significantly biased.

The confusion matrix for the maximum classification performance on this dataset is shown in Figure 7b. This matrix is immensely helpful in precisely illustrating the results of a categorization. Comparing the number of genuine samples for every class with the multitude of expected samples clearly illustrates the issue of misclassified samples. The numbers that appear diagonally from the upper left to lower right in a matrix indicate the count of accurately classified instances for each respective category. Any number that diverges from the diagonal demonstrates a classification error by the classifier. As seen in Figure 7b, the model faced some challenges while classifying the items of the pick, sit-up, and run classes. The misclassifications are marked in blue boxes in the confusion matrix. Despite some minor drawbacks, the model’s performance on the KU-HAR dataset was flawless.

### 3.4. Results on UCI-HAR and WISDM Dataset

The AM-DLFC model was engineered utilizing the KUHAR dataset with the goal of enhancing accuracy in comparison to prior models. However, to assess the model’s robustness, feasibility, and effectiveness, it was also evaluated on the UCI-HAR and WISDM datasets. The acquired accuracy, loss, and F1 curves attained after evaluating the datasets are depicted in Figure 8. Figure 8a,b demonstrate the outcomes of the accuracy obtained in all the epochs on both training and testing sets of the UCI-HAR and WISDM datasets, respectively. On the testing subgroup of the UCI-HAR dataset, the best accuracy was around 94.25% and was attained in the 195th epoch. The overall accuracy achieved was 93.48%. The corresponding loss curve and the F1 curve illustrated in Figure 8c,e, respectively, exhibit congruent results. Regarding the WISDM dataset, a carefully selected subset of 13,841 samples was utilized for training, with the remaining 25% of samples being reserved for testing. The overall and the highest accuracy achieved were 93.89% and 94.26%, respectively, as can be observed in Figure 8b.

Figure 9 shows the graphical depiction of the confusion matrix for the best classification result using the UCI-HAR and WISDM datasets. Figure 9a shows that the UCI-HAR model had some difficulty differentiating between samples from the classifications “Sitting”, “Standing”, and “Laying”.

As shown in Figure 9b, a thorough examination of the confusion matrix for the WISDM dataset revealed that several samples, which were originally classified as “Upstairs” and “Downstairs” activities, were inaccurately classified as “Walking” and “Jogging” activities. This discrepancy can be attributed to the fact that all four activities possess a shared characteristic in that they entail fundamental walking movements, albeit at varying velocities and along distinct planes. Despite this, the model displayed an almost impeccable capability in differentiating between the “Stand” and “Sit” samples. Nonetheless, the overall performance of the model on both datasets was deemed satisfactory.

### 3.5. Discussion

The outcome of the proposed AM-DLFC model was evaluated concerning recent datasets. The initial goal of the AM-DLFC model was to improve the recognition accuracy in terms of the KU-HAR dataset. The dataset is relatively new, and only a few research studies have been carried out with this dataset. A comparison of the accuracy of the KU-HAR dataset with the previous research is illustrated in Table 4. Despite the KU-HAR dataset being relatively new, the AM-DLFC model demonstrated superior performance when compared to the two previously published articles. Specifically, the proposed model exhibited 7.2% and 8.33% enhancement in overall classification accuracy when compared to the studies conducted by [19] and [34], respectively. In terms of the F1 score, the achieved score was enhanced by around 9.33%, 8.4%, 3.81%, and 2.67% compared to [19,34,35], respectively. Despite having a close accuracy result with [36], the AM-DLFC showed better performance overall. The proposed model achieved around 95% accuracy around the 60th epoch, which, in the case of [36], was around the 75th epoch roughly. Moreover, the AM-DLFC model reached over 90% accuracy around the 25th epoch, and in the case of [36], it was around the 30th epoch, which shows better stability. The proposed model also shows a peak accuracy of almost 97% which is higher than [36]. However, as seen in Figure 7, the AM-DLFC model faced some challenges while classifying the items of the pick, sit-up, and run classes, which suggests that there is potential for further advancements in this field utilizing the KU-HAR dataset. Furthermore, the AM-DLFC model was also evaluated on the UCI-HAR and WISDM datasets, which showed a modest accuracy of 93.48% and 93.89%, respectively. While these results may not reflect the highest scores compared to previous models, they demonstrate the proposed model’s robustness and viability as a solution for HAR.

## 4. Conclusions

This study introduces a DL model that incorporates an attention mechanism to extract more sophisticated features, leading to increased accuracy. Instead of utilizing complex signal processing techniques, the model utilizes the spectrogram of raw signals. The AM-DLFC model was evaluated on the KU-HAR dataset and demonstrated an impressive 96.86% classification accuracy and a peak accuracy of 97%. Additionally, the AM-DLFC model was further assessed on UCI-HAR and WISDM datasets to evaluate its robustness, yielding overall classification accuracy of 93.48% and 93.89%, with peak accuracy of 94.25% and 94.26%, respectively. The evaluation metrics further highlight the reliability and robustness of the proposed method compared to other existing works. This study reveals challenges in accurately classifying the pick, sit-up, and run activities using the AM-DLFC model within the KU-HAR dataset, suggesting a need for improvement in this specific area. Consequently, this finding highlights the potential for further advancements in accurately classifying the aforementioned activities. While the AM-DLFC model exhibited impressive performance on the KU-HAR, UCI-HAR, and WISDM datasets, its evaluation in real-world scenarios remains unexplored. Thus, the model’s effectiveness in practical settings, where external factors such as environmental noise and user variations can impact the signals, remains uncertain. In the later version of our work, we intend to implement our work in cutting-edge technologies (e.g., Raspberry Pi, Nvidia Jetson Nano) to evaluate the proposed approach in real-world settings. Although the proposed approach has demonstrated robustness and reliability, it is essential to acknowledge that existing HAR methods have achieved promising outcomes on the UCI-HAR dataset and WISDM dataset. Despite some difficulties in distinguishing between certain homogeneous activities, the model performed exceptionally well in recognizing most classes. It is hoped that this model will serve as a valuable resource in the development of intelligent applications for various smart devices.

## Figures and Tables

**Figure 1 sensors-23-05715-f001:**
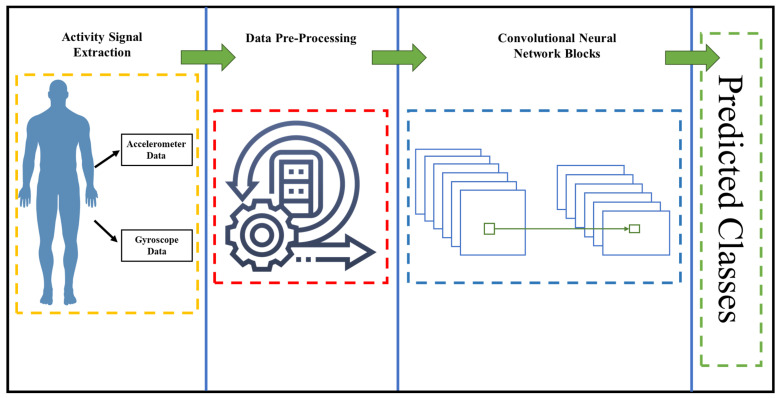
Methodology of the proposed model.

**Figure 2 sensors-23-05715-f002:**
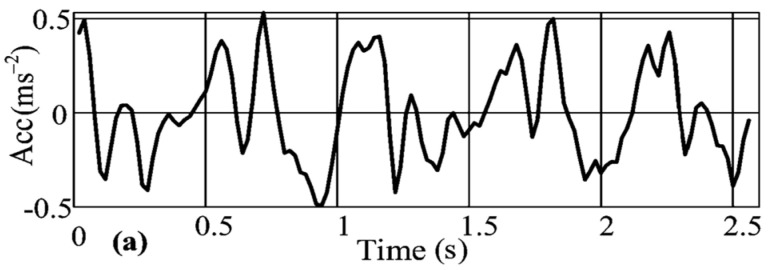
One-dimensional time-series signal.

**Figure 3 sensors-23-05715-f003:**
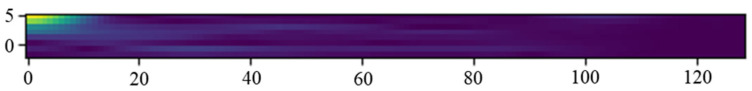
Converted spectrogram from the 1D time-series signal.

**Figure 4 sensors-23-05715-f004:**
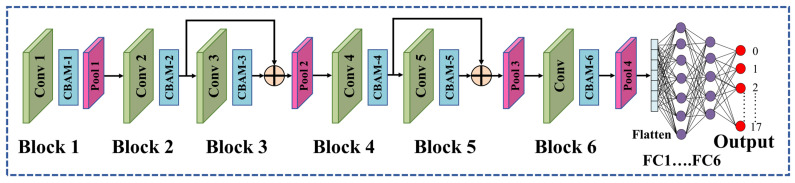
The architecture of the developed AM-DLFC model (this figure shows 18 neurons in the output layer representing the KU-HAR dataset). Our code is available at https://github.com/Masrur02/AM_DLFC (accessed on 13 June 2013).

**Figure 5 sensors-23-05715-f005:**
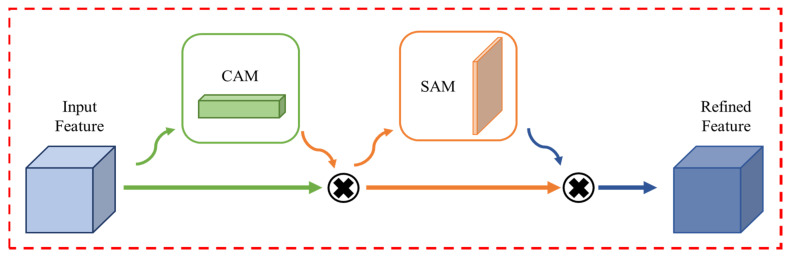
CBAM module structure.

**Figure 6 sensors-23-05715-f006:**
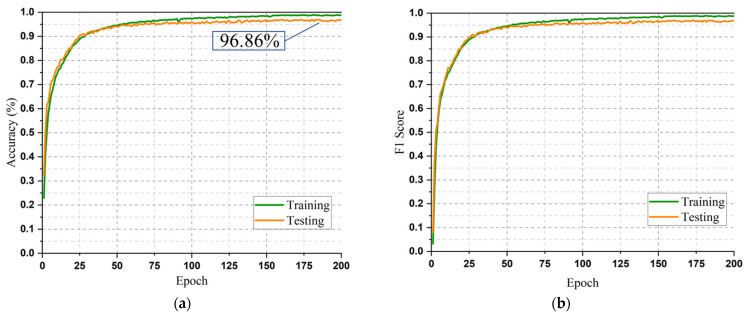
Classification accuracy, loss, and F1 curves achieved from the experiment on the KU-HAR dataset. (**a**) Accuracy; (**b**) F1 Score.

**Figure 7 sensors-23-05715-f007:**
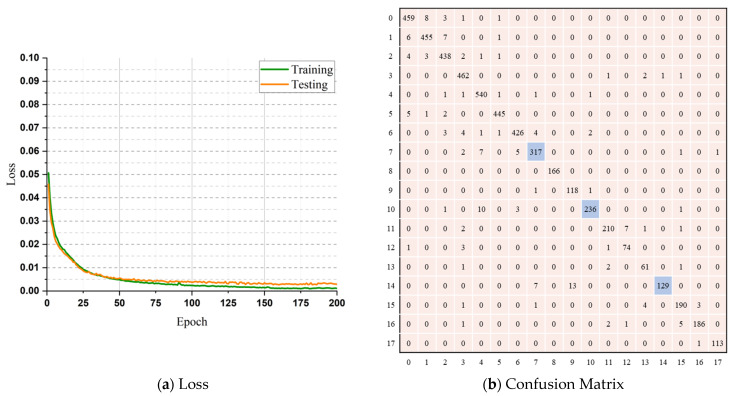
The loss function and the confusion matrix of the KU-HAR dataset. (**a**) Loss; (**b**) Confusion Matrix.

**Figure 8 sensors-23-05715-f008:**
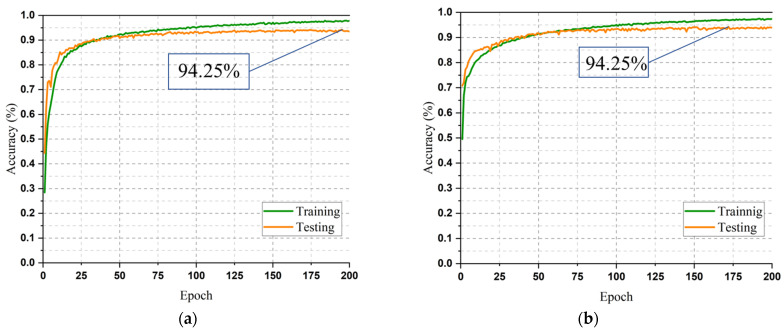
Classification accuracy, loss, and F1 curves achieved from the experiment on UCI-HAR and WISDM datasets. (**a**) UCI-HAR dataset accuracy; (**b**) WISDM dataset accuracy; (**c**) UCI-HAR dataset loss; (**d**) WISDM dataset loss; (**e**) UCI-HAR dataset F1 score; (**f**) WISDM dataset F1 score.

**Figure 9 sensors-23-05715-f009:**
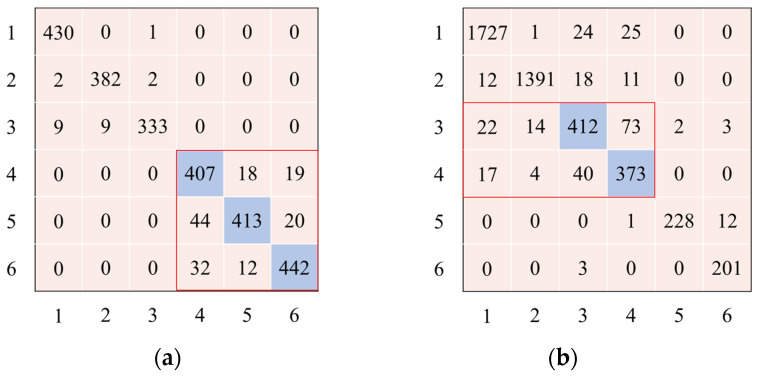
The confusion matrix for UCI-HAR and WISDM datasets. (**a**) UCI-HAR dataset confusion matrix; (**b**) WISMD dataset confusion matrix.

**Table 1 sensors-23-05715-t001:** KU-HAR dataset activities.

Class Name	Class ID	Performed Activity	Duration or Repetitions per Sample
Stand	0	Standing still on the floor	1 min
Sit	1	Sitting still on a chair	1 min
Talk–sit	2	Talking with hand movements while sitting on a chair	1 min
Talk–stand	3	Talking with hand movements while standing up or sometimes walking around within a small area	1 min
Stand–sit	4	Repeatedly standing up and sitting down	5 times
Lay	5	Laying still on a plain surface (a table)	1 min
Lay–stand	6	Repeatedly standing up and laying down	5 times
Pick	7	Picking up an object from the floor by bending down	10 times
Jump	8	Jumping repeatedly on a spot	10 times
Push-up	9	Performing full push-ups with a wide-hand position	5 times
Sit-up	10	Performing sit-ups with straight legs on a plain surface	5 times
Walk	11	Walking 20 m at a normal pace	≈12 s
Walk backward	12	Walking backward for 20 m at a normal pace	≈20 s
Walk-circle	13	Walking at a normal pace along a circular path	≈20 s
Run	14	Running 20 m at a high speed	≈7 s
Stair-up	15	Ascending on a set of stairs at a normal pace	≈1 min
Stair-down	16	Descending from a set of stairs at a normal pace	≈50 s
Table tennis	17	Playing table tennis	1 min

**Table 2 sensors-23-05715-t002:** UCI-HAR dataset activities.

Activity	Description	No. of Samples
Walking	Participant walks horizontally forward in a direct position	1722
Walking (Upstairs)	Participant walks upstairs	1544
Walking (Downstairs)	Participant walks downstairs	1406
Sitting	Participant sits on a chair	1777
Standing	Participant stands inactive	1906
Laying	Participant sleeps or lies down	1944

**Table 3 sensors-23-05715-t003:** WISDM dataset activities.

	Raw Data	Transformed Data
Samples	1,098,207	5424
Attributes	6	46
**Class Distribution**		
Walking	38.60%	38.40%
Jogging	31.20%	30.00%
Upstairs	11.20%	11.70%
Downstairs	9.10%	9.80%
Sitting	5.50%	5.70%
Standing	4.40%	4.60%

**Table 4 sensors-23-05715-t004:** AM-DLFC model performance comparison with existing models using KU-HAR dataset.

KU-HAR		
Research	Accuracy (%)	F1 Score (%)
[19]	89.67	87.59
[36]	96.67 (Peak)	96.41 (Peak)
[34]	88.53	88.52
[35]	Not Provided	94.25
**Proposed**	**96.86**	**96.92**

## Data Availability

Not applicable.

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
