# Peer review of "Human Activity Recognition Using Attention-Mechanism-Based Deep Learning Feature Combination"

_sensors, 2023, doi:10.3390/s23125715_

Round 1

Reviewer 1 Report (Previous Reviewer 4)

Thanks for addressing my questions. 

Author Response

Reviewer 2 Report (New Reviewer)

A very good job, and the updated parts are very readable.

1. Major.

The authors used a fixed-length window.

1.1 The authors do not explain 1) why the duration is fixed at 3 seconds and 2) why there is no overlap. This can be clarified by simple and reasonable explanations, such as comparing the results of different window lengths and overlap and taking the best value. 

1.2 One risk that comes with 3 seconds is the real-time HAR application. Early literature shows that good offline and real-time recognition rates are achieved with 400ms windows and 50% overlap (BIODEVICES 2019 Best Paper), also based on a dataset with a large variety of activities, using predominantly inertial sensors, with or without EMG/goniometer, which is a near-perfect accuracy-latency tradeoff. Also, compared to ML+handcrafted features, like "on a real real-time wearable human activity recognition system," DL brings much more computational complexity, further affecting real-time performance. BTW, an HAR system with a 3-second window plus DL decoding time is weak to produce a good interaction, control, or entertainment experience (of course, this is not what you need to pay attention to at present).

1.3 Also noteworthy is the natural duration of human daily activities. Recent studies have shown that all 22 investigated human single motions (e.g., sitting down, standing up, one gait in walking/jogging/upstairs/downstairs/shuffling, single-leg jumping, double-leg jumping) last in the interval of 1–2 seconds, and the duration of each activity is normally distributed among the healthy population – "How Long Are Various Types of Daily Activities." Thus, it seems that 3 seconds may include one-and-a-half-stride data or, for other recognizable motions, redundant signals. Since your model is based on spectral characteristics, a longer window may bridge the leaking effect of windowing, but still, the time of 3 seconds is a question mark to explain.

Minor.

2. I also agree that using the deep representation of features is to some extent more convenient than handcrafted. But this should not be overemphasized to reflect the advantages of this paper.

2.1 The literature in 2022, also on Sensors, shows that HAR based on handcrafted features is not globally inferior to DL overall: https://www.mdpi.com/1424-8220/22/19/7324.

Do more literature research. 

2.2 ML is not only about traditional models like kNN, RF, and SVM but also includes modeling methods like HMM that are naturally suited to the time series properties of HAR ("hidden Markov model and its application in human activity recognition and fall detection"). In most of the comprehensive literature, HMM, even with handcrafted features, is definitely not inferior to DL in terms of recognition rate and is usually superior in terms of efficiency and simplicity.

2.3 "The commonly used feature extractors are often dataset-specific, limiting their generalization to other datasets and objectives." is not agreeable. First, the literature mentioned in 2.1 studied the domain generalization of handcrafted features. An additional, essential instance, HMM, as mentioned in 2.2, with its latest advanced "motion units Generalized sequence modeling," has a wide range of generalizability, scalability, and universality. More notably, according to the literature, it has no recognition rate disadvantage compared to DL, but has better interpretability and modeling efficiency/simplicity. For instance, in real-time, even interactive HAR scenarios ("interactive and interpretable online human activity recognition"). For the "generalization" you mentioned, the types of manual features may not be completely uniform for different datasets, but works like "feature-based information retrieval of multimodal biosignals" and "biosignal processing and activity modeling for multimodal human activity recognition (Table 5.8)"  provide the most significant ~30 features with low computation cost from a comprehensive TSFEL feature bank of 60 features in time, statistical and frequency domains. These features have been repeatedly (over dozens of times) shown to be appropriate and effective for human activity modeling and recognition on various datasets, already addressing the "generalization" problem you mentioned. Even if all of the 30 features are used, the computational efficiency should still be no worse than that of DL. The latter literature also provides a framework for feature selection, including greedy, ANOVA, MRMR, etc.

The features automatically learned by deep representations are certainly worth investigating for HAR. But as seen above, the reason should not be the generalization, simplicity, or recognition rate of applying handcrafted features. They do not have natural disadvantages as you mentioned.

There are still some grammatical flaws and typos, several on each page,

Some examples:

165 withe THE same number of

184 three datasetS'

Round 2

Reviewer 2 Report (New Reviewer)

I note that:

1. [35] is a very early (2015) article and does not cover many of the new HMM-based HAR technologies. For example, the improved HHMM (hierarchical HMM) and Motion Units (Generalized Sequence Modeling of human activities for sensor-based activity recognition) are not included. A suggested citation in this direction is https://doi.org/10.1007/978-981-19-0390-8_108 in 2022. Meanwhile, the latest platform and interactive machine learning solution in hand-crafted feature-based HAR should be given in lines 74-76: https:// doi.org/10.1109/PerComWorkshops53856.2022.9767207.

2. In the response, the authors fixed only two expression errors for which I gave examples, and as I pointed out in the first round, the article also contains many English expression errors in each page. The expression quality of the current version is not OK for the publication. Please double-check and proofread.

See above.

Author Response

This manuscript is a resubmission of an earlier submission. The following is a list of the peer review reports and author responses from that submission.

Round 1

Reviewer 1 Report

Discussion part need to be improved as well as conclusions part. More references from 2022, 2023 especially from Q1-web of science SCI indexed journals expected. Especially from Pattern recognition and deep learning specific journals [IEEE transactions and ACM transactions expected].

Modify the paper and resubmit for further suggestions.

Check the paper using grammarly. Few errors found.

Reviewer 2 Report

This work presents a novel HAR classification method using a functional DL model which is integrated with an attention module to extract the crucial characteristics from sensor data and obtain higher classification accuracy.

The paper discusses a hot topic of the related literature that the reader of this Journal would like to read.

While this is a very interesting paper, I think it is necessary to address some concerns before publication.

Some improvements should be done for a better comprehensive reading.
I would suggest to the authors that include some discussion about explainability for the results. Also, the following issues should be improved:

1.         In the abstract, the novelty of this research should be discussed.

2.         In the introduction, the motivation and contribution of this paper should be given.

3.         Since the literature review is quite poor and to support several assertions, the authors are adivsed to use the following references:

             a. L. Dhammi and P. Tewari, "Classification of Human Activities using data captured through a smartphone using deep learning techniques," 2021 3rd International Conference on Signal Processing and Communication (ICPSC), Coimbatore, India, 2021, pp. 689-694, doi: 10.1109/ICSPC51351.2021.9451772.

          b. Valagkouti, I.A.; Troussas, C.; Krouska, A.; Feidakis, M.; Sgouropoulou, C. Emotion Recognition in Human–Robot Interaction Using the NAO Robot. Computers 2022, 11, 72. https://doi.org/10.3390/computers11050072.

       c. P. Jantawong, A. Jitpattanakul and S. Mekruksavanich, "Enhancement of Human Complex Activity Recognition using Wearable Sensors Data with InceptionTime Network," 2021 2nd International Conference on Big Data Analytics and Practices (IBDAP), Bangkok, Thailand, 2021, pp. 12-16, doi: 10.1109/IBDAP52511.2021.9552133.

4.         A schema describing the research methodology could be helpful.

5.         More information about the research methodology could be included.

6.         The conclusions should lead to new knowledge. Also, limitations of this research are missing at the moment.

Concluding, the structure of paper is good, but the main contributions of the paper do not add significant value to the existing body of knowledge in the related subject area.
A suggested contribution is to have a discussion section to compare the presented work with the related work in the literature.

Overall, the paper is well organized. However, it lacks critical discussion in contrast with the related work in the literature and does not provide major contributions to the field.

Reviewer 3 Report

In this manuscript, the authors propose an approach called ‘Human Activity Recognition Using Attention Mechanism Based Deep Learning Feature Combination’, which mainly focuses on attention mechanism and feature combination.

 1.   There needs to be more content in the introduction about the proposed method and the analysis part of the experimental results to enhance the credibility of the article. 

2.   The proposed approach was evaluated on a single dataset (KU-HAR), which may limit its generalizability to other datasets. 

3.   The study did not compare the proposed approach with other state-of-the-rt methods for human activity recognition, which may limit its ability to demonstrate superiority over different approaches.

4.   The study should have evaluated the proposed approach in real-world settings, which may limit its practical applicability.

5.   To ensure that the text flows smoothly and has a clear cause-and-effect relationship between sentences, it is recommended to put datasets details in the experiment part. This will help to improve the overall quality of the article.

6.   The current HAR methods have achieved promising results on other datasets.  

Reviewer 4 Report

Summary

The study proposed a DL model with feedforward layers for HAR applications. Important details about datasets, data preprocessing and how the proposed model works for the 3 datasets is missing. The top challenge of using sensor data for HAR application is to address individual differences; normally cross-subject validation is critical. The evaluation method done by this work is very simple that only a random split on training and testing is conducted, even without a validation set. The comparison to other studies is misleading as the results of other studies are from different evaluation methods. The improvement of recognition accuracy does not seem convincing if the results are reported on the test set. The writing needs to be very much improved.

Minor points

1.      It is very questionable about the statement in the abstract: “However, most of the existing works do not explore the state-of-the-art modules of DL models and modify them to obtain higher accuracy and adaptability.” HAR has been a hot topic for over 20 years. There are so many research studies exploring the DL methods which were applied to different sensor data of different activities. What are “state-of-the-art DL modules” you mean here and what they are used for?

2.      Since this study focuses on using sensor data from smart phones, it is confusing to review the image-based methods in detail at the beginning of the introduction.

3.      The introduction needs to be rewritten to focus on the existing DL methods of using accelerometer and/or gyroscope data for HAR applications and explain what the “state-of-the-art DL modules” are. The challenges of HAR should also be specified.

4.      A DL model with 2.5M parameters does not seem lightweighted. The TinyML models used on microcontrollers are much smaller. It would be clearer to compare the model size to the existing DL models applied to similar datasets.

5.      The activities in Figure 1 do not match to any of the dataset, which makes it confusing. It is suggested to list the activities included in each dataset in Section 2. It is confusing why there are 18 classes/outputs of the proposed model.

6.      What is the sensor data input to your model? Some datasets have 3-axis accelerometer data and 3-axis gyroscope data, while some only have 3-axis accelerometer data. Was each axis of the sensor data converted to spectrum? Or the sensor data was preprocessed before converting to spectrum? It is suggested to add the details of data preprocessing for each dataset.

7.      It states the proposed model has 6 convolutional blocks, but only 5 blocks are shown in Figure 4.

8.      Does the same proposed model structure work for all three datasets, but only parameters need to be tuned? It states the input to the model is with the shape of 8×129×6, but not clear what the data layout is. It is suggested to clearly state the input data dimensions of each dataset and if the input layer needs to be changed according to input data dimension. If the same model structure works for three different datasets with small tuning, I think “generalized model structure with CBAM modules” seem to be the innovation of this study and needs to be specified at the beginning along with the challenge that this generalized model structure addressed.

9.      In the evaluation and result section, it sounds like the results are only from training with 75% data and tested with 25% data. The result of this paper was compared to other studies in Table 1. If k-fold cross-validation was not done with this study, the comparison in Table 1 is misleading, as the results of some studies in Table 1 were for k-fold cross-validation. It is suggested to use the same evaluation method as the other studies and clearly state what the evaluation methods are used by other studies.

English needs to be improved. Some descriptions are not clear. There are grammar errors throughout the manuscript.